# Instability of estimation results based on caliper matching with propensity scores

**Kazushi Maruo**[ID][1]*, **Yusuke Yamaguchi**[2], **Ryota Ishii**[1], **Masahiko Gosho**[1]

1 Department of Biostatistics, Institute of Medicine, University of Tsukuba, Tsukuba, Ibaraki, Japan,
2 Biostatistics, Data Science, Astellas Pharma Global Development, Inc., Northbrook, Illinois, United States of America

* maruo@md.tsukuba.ac.jp

## Abstract

Caliper matching is often used to adjust for confounding biases in observational studies. This method with random order matching allows for the cherry-picking of the analysis results to suit the analyst's convenience. Random order matching can also result in large fluctuations in the analysis results due to small additions and/or changes in data. These "instability problems" might compromise the reproducibility of the study results. Some studies have discussed instability issues, but the conditions are limited, and there is no knowledge of which alternative order method should be used instead of the random order method. We evaluate the instability problem by calculating the extent to which the results can vary within a single study dataset and provide guidelines for choosing the best alternative matching order method based on simulations and a case study. From simulation studies, instability might be serious when the sample size was small, the true odds ratio was large, the proportion for the treatment group was large, and the c-statistic for the propensity score model was large. We recommend not using random order matching and instead using lowest to highest score order matching or the median of multiple random order matching results. We also recommend pre-specifying the matching order method.

## 1. Introduction

One of the most important challenges in non-randomized observational studies is adjusting for confounding bias. Propensity score analyses [1,2] are frequently used to reduce confounding bias. The propensity score represents the probability of treatment assignment conditional on observed baseline covariates. There are four methods in which the propensity score can be used: matching on the propensity score, stratification on the propensity score, covariate adjustment using the propensity score, and inverse probability of treatment weighting using the propensity score [2,3]. Among them, propensity score matching is frequently used in medical and social research articles because of its ease of interpretation and implementation for clinicians [3–6] despite the issue of reduced sample size due to matching.

**Data availability statement:** The data set used in Case study is available from the R package, mlr3proba [22] (https://mlr3proba.mlr-org.com/index.html).

**Funding:** This work was supported by Japan Society for the Promotion of Science (JSPS) KAKENHI Grant Numbers JP22K19682 and JP24K02662. The funders had no role in study design, data collection and analysis, decision to publish, or preparation of the manuscript.

**Competing interests:** The authors have declared that no competing interests exist.

Austin [3] compares several algorithms for matching on the propensity score using statistical simulations, where greedy nearest neighbor matching without replacement within specified caliper widths shows best performances regarding mean squared error for treatment effect. This method is often simply referred to as "caliper matching". In the caliper matching, the estimation result of the treatment effect depends on the order of matching. Austin [3] concludes that the selection on order of matching had, at most, a modest effect on estimation. This is true from the standpoint of accuracy and precision. Turning to a single dataset, however, changing the order of matching could significantly alter the estimation results, especially for small-to-medium sample sizes. For example, the seed can be cherry-picked as to be convenient for the research objectives when the random order matching is employed. In this article, this problem is referred to as the instability for analysis results. If the analysis methods, including the seeds of random numbers, are fully pre-specified in the protocol or statistical analysis plan, the instability problem does not arise. However, in non-randomized observational research, such a pre-specification is often not performed, and unfortunately, cherry-picking of results is rather easy to conduct. Furthermore, although the instability problem may not be visible, different researchers may set different seeds, which may lead to different results despite the full prespecification even if they used the same data set.

In observational studies, multiple analyses may be performed as data accumulates. For example, the data may be analyzed every few years as the database is accumulated. Another example is that in observational studies where the statistical analysis plan is not strict, data may be added and/or changed after the statistical analysis. In this study, we consider the latter case where small data additions occur. In such cases, even if the seed is fixed, the matching order may change significantly between analyses. This might cause changes in estimates that greatly exceed the variation due to the accumulation of additional data. This problem is also part of the instability problem.

Komen et al. [7] evaluated this instability problem with simulation and case studies. They concluded that random order matching should not be used, but no studies have examined optimality in the sense of instability among alternative non-random order matching methods. Komen et al. [7] also conducted their simulations assuming epidemiological studies with moderate to large sample sizes, but no studies have been evaluated in settings assuming small to medium observational clinical studies, which would be more influenced by instability. Specifically, we evaluate the degree to which the analysis results may vary using random order matching, that is, the instability of the analysis results, under various situations. This instability reflects the range of analysis results that can be cherry-picked. Furthermore, no studies have examined the variability in propensity score analyses arising from multiple analyses associated with data accumulation. In this paper, we conduct simulation studies to provide answers to these unsolved problems. We primarily focused on the odds ratio based on logistic regression as an effect measure. Although this odds ratio is a non-collapsible effect measure (e.g., see Whitcomb and Naimi [8]) that requires careful interpretation, it remains widely used and is the primary focus of this study.

In summary, this study has two primary objectives: first, to evaluate the instability of matching procedures in observational studies with small to moderate sample sizes and binary outcomes; and second, to provide practical guidelines for selecting the most appropriate matching method. These objectives were addressed through simulation studies and a case study designed to reflect the characteristics of the problem setting. The simulation settings were carefully constructed to capture a wide spectrum of real-world scenarios. The alerts and recommendations presented in this study aim to assist readers in selecting an appropriate matching procedure with respect to instability.

The remainder of this paper is organized as follows: Section 2 provides an overview of propensity score matching, Section 3 describes the simulation study, Section 4 presents a case study, and Section 5 concludes the study.

## 2. Greedy nearest neighbor matching

This study focuses on greedy nearest neighbor matching with calipers, which is also simply called caliper matching. Austin [3] showed, based on simulations, that caliper matching performs better in the context of least-squares error than other methods, such as greedy nearest neighbor matching without calipers, matching with replacements, and optimal matching. Greedy nearest neighbor matching first selects a treated subject and identifies the untreated subject with the closest propensity score as a matched control subject. In the caliper matching, treated and untreated subjects are matched only if the absolute difference in their propensity scores is within a prespecified maximal distance (the caliper distance). Because greedy nearest neighbor matching does not uniquely determine the order in which the treated subjects are selected, the order must be determined in some way. Austin [3] compared four approaches for selection order: 1. treated subjects were selected sequentially from highest to lowest propensity score; 2. from lowest to highest score; 3. the order of the best possible match, that is, the selected treated subject is the subject who is closest to a control subject; 4. random order. The performance of these four approaches is almost the same in terms of the mean-squares error [3].

Random order matching suffers from the problem of cherry-picking through seed selection. The methods 1–3 do not have such a problem, but the problem of variability remains, where the matching pairs may change owing to data accumulation. Furthermore, in Method 1, participants with higher propensity scores in the treatment group were more likely to be matched with closer participants in the control group. That is, the confounding bias in the higher-scoring population was adjusted for priority. For Method 2, the opposite is true. Method 3 showed no such tendency.

Let us consider the influence of the matching algorithm on repeated analyses conducted through data accumulation. Given that the propensity scores across successive datasets are correlated, the probability of the same pairs being matched in each analysis is relatively high for Methods 1–3 as they use deterministic matching procedures (e.g., Method 1 prioritizes pairing units with the highest propensity scores). In contrast, random matching reduces the probability of such overlap. This raises concerns about potential instability in the results across repeated analyses.

These ordering methods are available in the MatchIt package [9], which is one of the most frequently used R software packages for matching. Methods 1, 2, 3, and 4 can be implemented by specifying "largest," "smallest," "closest," and "random" in the m.order argument of the matchit function, respectively. One possible solution to these problems is to estimate treatment effects based on multiple random matchings and to estimate the median of multiple estimates. Odds ratios may increase significantly when the denominator odds are close to zero. However, even if a small number of extreme estimates are included among multiple random matches, the median remains less sensitive to them.

## 3. Simulation

### 3.1. Simulation design

This section presents the design of the simulations performed to evaluate the degree of instability caused by the matching order.

First, we describe the propensity score model as $logit(p_{gj}) = \beta_{gi} + \beta_{gx}X_{cj}$, where $p_{gj}$ is the probability of being allocated to the treatment group for subject $j(j = 1, \cdots, N)$, $N$ is the total sample size, $\beta_{gi}$ and $\beta_{gx}$ are regression

coefficients for the propensity score model, and $X_{cj}$ is a covariate. $X_{cj}$ followed a standard normal distribution, standardized multinomial distribution with 10 categories ($\boldsymbol{p} = (1, 2, 3, 4, 5, 5, 4, 3, 2, 1)^T/30$), or standardized mixture normal distribution with the probability density function $f_x = \frac{1}{2}\phi\left(x; -2\tau, \tau^2\right) + \frac{1}{2}\phi\left(x; 2\tau, \tau^2\right)$, where $\phi$ is the probability density function of normal distribution and $\tau$ is standardization parameter. These distributions are denoted unimodal, categorical, and multimodal, respectively. $\beta_{gi}$ and $\beta_{gx}$ were set so that $p_g = E\left(p_{gj}\right)$ and the c-statistic as a measure of the "fit" of the propensity score model were set to specific values. $p_g$ was set as 0.1, 0.2, or 0.4. A sample size of the treatment group is generally smaller when conducting matching, and these three values were set to cover as wide a range as possible with $p_g < 0.5$. The c-statistic for the propensity score model was set as 0.6 or 0.85. The c-statistic ranges from 0.5 to 1. If it is considerably close to 0.5, it is equivalent to a randomized controlled trial, and if it is substantially close to 1, the treatment selection is deterministic, which may make any confounding adjustment less meaningful. Therefore, the c-statistics of 0.6 and 0.85 were selected as cases of weak and strong confounding influence, respectively, within realistic and useful ranges. Fig 1 shows a histogram of the propensity score from 100,000 random numbers for different proportions of treated subjects ($p_g$) and the c-statistic for the propensity score model for the 3 types of covariates. It may appear unrealistic to include only one covariate in a propensity score model. However, in estimating the propensity score model, we are less concerned with the effect of each covariate, and the distribution of the estimated propensity score itself will significantly affect the results. In this sense, our setting assumed that the distribution of propensity scores was unimodal, categorical, or multimodal (bimodal), which does not appear to be an unrealistic assumption. This setting can also be considered as a linear combination of multiple covariates. A unimodal or multimodal distribution represents a linear combination of a variable set that includes continuous variables, whereas a multinomial distribution represents a linear combination of categorical variables only, which was set with the intention of generating a tied propensity score. Based on this model, the group variable $X_{gj}$, was generated using the Bernoulli distribution $X_{gj} \sim Ber\left(p_{gj}\right)$, where $X_{gj} = 0$ (control group) and 1 (treated group).

Subsequently, we describe the outcome model: $logit(p_{ej}) = \beta_{ei} + \log(OR)X_{gj} + X_{cj}$, where $p_{ej}$ is the event occurrence probability for subject $j$, $\beta_{ei}$ is an intercept, and $OR$ is the odds ratio, that is the treatment effect. $\beta_{ei}$ was set such that $p_e = E\left(p_{ej}\right)$ was a specific value for a given value of $OR$. $p_e$ was set as 0.1, 0.3, or 0.5. To interpret the odds ratio based on logistic regression as a risk ratio, $p_e$ must be close to 0, thus, we set $p_e \leq 0.5$. Note that when $p_e = 0.5$, it is difficult to interpret the odds ratio as a risk ratio. If $p_e$ is significantly small, the number of events may be insufficient for matching, making it infeasible. Therefore, we set the lower limit of $p_e$ to 0.1. $OR$ was set as 0.5, 0.75, or 1, which indicated that the treatment reduced ($OR = 0.5$: large effect, 0.7: medium effect) or did not change ($OR = 1$) the risk of event occurrence. Based on this model, the outcome variable $Y_j$, was generated using the Bernoulli distribution $Y_j \sim Ber\left(p_{ej}\right)$, where $Y_j = 0$ (no event) or 1 (event).

The total sample size $N$, was set to 100, 200, or 400. These settings were intended to simulate a relatively small-sample situation as a nonrandomized study. Especially when $N$ = 100, $p_e = 0.1$, and $p_g = 0.1$, the expected number of events was one for each group, even in the no-caliper case, indicating a very small information setting. In such scenarios, the asymptotic assumptions underlying statistical inference are entirely violated. This setting was deliberately designed to highlight the potential pitfalls of applying matching methods under conditions of extreme data sparsity. The number of simulation settings was 3 (type of covariate) × 3 ($p_g$) × 2 (c statistc) ×3 ($p_e$) × 3 ($OR$) × 3 ($N$)= 486. The number of simulations for each setting was 1,000. In each simulation, we estimated the odds ratio with greedy nearest neighbor matching with a caliper. The procedures for the matching order were random (M_Rand), highest to lowest (M_HtoL), lowest to highest (M_LtoH), or closest match (M_Clos). Matching without a caliper in a random order (M_NoCal) was also performed. Matching was conducted with the logit of the propensity score, and the caliper was set at 25% of the standard deviation of the logit of the propensity score (e.g., see Rosenbaum and Rubin [10]). A caliper of 15% was also set for reference. Matching ratios were set for two situations: 1:1 and 1:2 for the treatment group to control group.

As a reference, we performed propensity score-weighted analyses for comparison. We used the average treatment effect for treated (ATT) weight (e.g., see Guo and Fraser [11]) as the estimate (W_ATT). As a modified version of W_ATT,

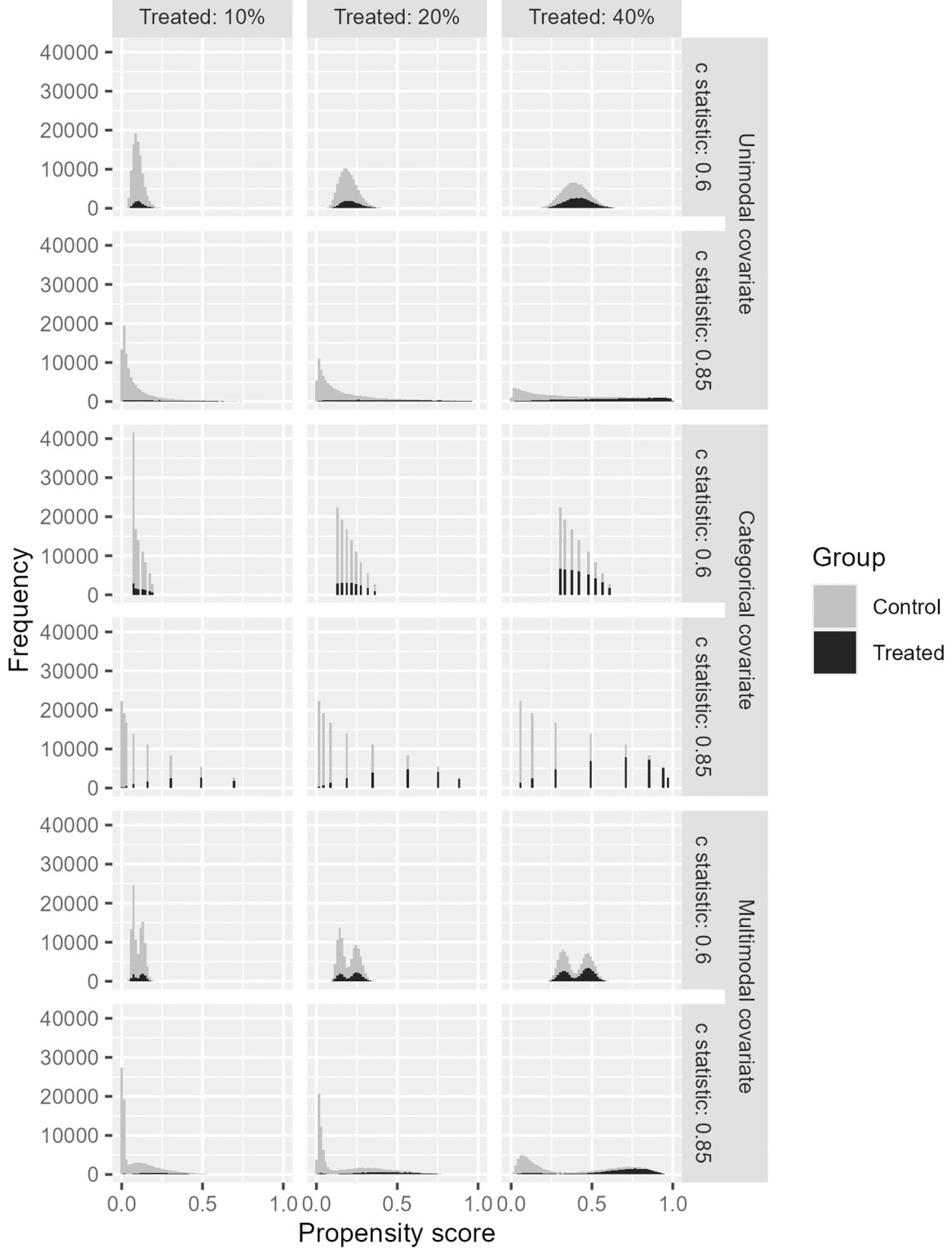

**Fig 1. Histogram of propensity score from 100,000 random numbers for different values of proportion of treated subjects and the c statistic for the propensity score model for 3 types of covariate.**

a method that excludes more than the 95th percentile of the distribution of propensity scores was also applied (W_ATTt) (Lee *et al.* [12]). Matching using calipers often excludes a portion of the treated group and estimates the treatment effect in the overlapping populations of both groups. Therefore, we also performed analyses based on the overlap weights (W_ATO) [13,14]. In the analysis for the random order (M_Rand, M_NoCal), the odds ratio estimation was repeated 100 times for one data set, and the width (97.5 percentile-2.5 percentile) of these 100 estimates was calculated. The median width was calculated for each simulation setting. We also calculated the median of 100 estimates for M_Rand, which is considered a new estimate of the odds ratio and is described as M_Med.

For each method (M_Rand, M_HtoL, M_LtoH, M_Clos, M_NoCal, W_ATT, W_ATTt, W_ATO, and M_Med), we calculated the median bias and interquartile range (IQR) of the odds ratios, where IQR was regarded as the index for precision. We added $0.1N$ (10%) of the data to each simulation, and the absolute difference between the odds ratio of the added merged data and that of the original data and the median of the absolute difference were calculated. This data aggregation process was designed to evaluate the instability of the analysis results when multiple analyses are performed as data accumulate in the same study, as described in the Introduction. We also estimated model-based and robust confidence intervals (CI) for the odds ratios in each simulation. Robust confidence intervals were estimated by using "CR2" option [15,16] in the type argument of the vcovCR function of the clubSandwich package [17] in R. The model-based and robust standard errors (SEs) for the log odds ratio were also estimated. For the M_Med method, CIs and SEs were calculated on matched data such that the OR equaled the median of 100 estimates and were calculated with the modified multiple outputation technique [18]. The latter is denoted as M_Med2. The proportion that CIs covered the true value in 1000 simulations for each setting was calculated as an estimate of the coverage probability (CP) of the confidence interval.

### 3.2. Simulation results

First, we describe the results of the median width for random order matching. For the method without a caliper (M_NoCal), figures are not included in the main text because of the large treatment effect bias. In addition, we included only the results for unimodal continuous covariates, 25% caliper, and 1:1 matching in the main text. See the Supplementary Material (S1 File (results for matching with caliper width of 25%, matching without caliper, weighting methods) and S2 File (results for matching with caliper width of 15%)) for all simulation results. Fig 2 illustrates the median width of the OR for 100 replications of random order matching (unimodal continuous covariates, 25% caliper, 1:1 matching). The median width was larger when the sample size was small, the true odds ratio was large, the proportion of subjects in the treatment group was large, and the c-statistic was large. However, the median width was smaller when the number of participants in the treatment group was small. The largest median width was as large as 1.25, which means that there is a greater than 50% chance that the preferred odds ratio can be selected from a range of 1.25 if cherry-picking of analysis results is performed. For covariate type, the multimodal covariates exhibited a slightly wider width at higher proportions of the treatment group and a narrower width at lower proportions, compared to the other types; however, the overall trend remained consistent across all covariate types. (see S1 and S2 Files in the Supplementary materials). In the 1:2 matching case, the median width was relatively wider than that in the 1:1 case; however, the trend was similar (S1 and S2 Files). In the absence of calipers, the median width was considerably narrower than that of the caliper matching, resulting in overlapping lines for each condition in the graph (S1 File). The width of the caliper (15% or 25%) had little effect on the results (S1 and S2 Files).

Fig 3 depicts the median bias for the OR (1:1 matching). Results are shown for cases with a c statistic of 0.85 and a true odds ratio of 1, such that the median width was the largest. For the weighting-based analysis method, only the results of the ATT weight-based method are presented in the text because it is one of the most frequently used methods. The W_ATT and M_HtoL methods had a slight upward bias, and the M_LtoH method had a slight downward bias; this tendency was more apparent when the proportion of the treated group was high. The bias for M_NoCal was large, and the bias for W_ATTt was slightly smaller than that for W_ATT. The M_Rand, M_Clos, and M_Med methods showed little bias, except

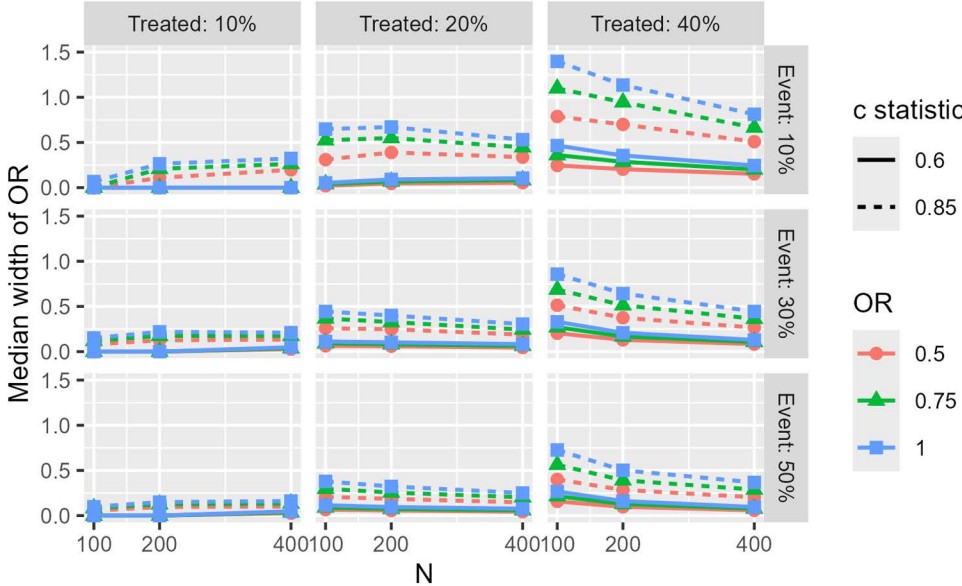

**Fig 2. Median width of estimated odds ratios for 100 replication of random order matching (unimodal continuous covariate, matching ratio: 1:1, caliper width: 25%).**

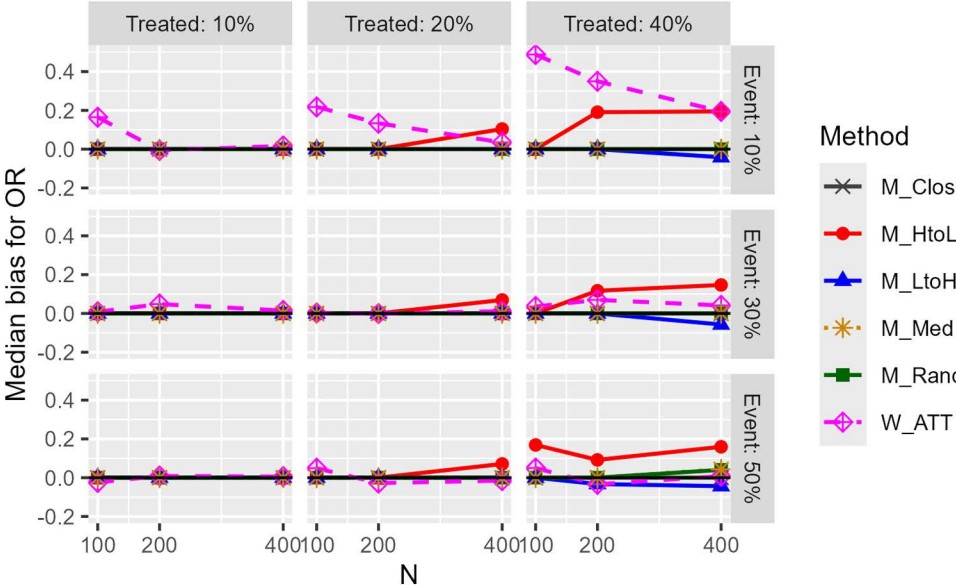

**Fig 3. Median bias for estimated odds ratios (unimodal continuous covariate, c statistic: 0.85, true OR: 1.0, matching ratio: 1:1).**

when the number of matched pairs was very small. The narrower caliper width (15%) had a slightly smaller bias than the wider case. The bias in the categorical covariate was considerably smaller than in the other covariate types. In the case of 1:2 matching, the bias was larger than in the case of 1:1 matching, particularly when the proportion of the treatment group was higher. However, the 1:2 matching would be less likely to be used when the proportion of treatment groups is high.

Although the matching ratio is not related to the weighting method, the results of the weighting method are also provided for both ratios as a reference.

Fig 4 shows the IQR of OR (precision; 1:1 matching). There was little difference between the matching methods, with W_ATT being inferior in some cases and superior in others. In some cases, the W_ATT method was more precise when the c-statistic was low. The order of precision of the weighting methods was W_ATO > W_ATTt > W_ATT. While some categorical covariates occasionally exhibited higher IQRs, overall differences across covariate types were minimal. In some cases, the narrower caliper (15%) exhibited slightly higher IQRs compared to wider one (25%). The results in the case of 1:2 matching were similar to those of 1:1 matching.

Fig 5 illustrates the results of the median absolute difference in OR for a 10% data addition (1:1 matching). The absolute differences in M_Rand were significantly greater than those of the other methods. The absolute difference of the M_HtoL method was slightly larger than that of the other methods. Overall, the absolute differences for the weighted method were smaller than those for the matching method, except when the sample size was small. The multimodal covariate setting showed more variation, but the order of variation in the matching procedure remained the same. The caliper width had little effect on the results. The results for 1:2 matching were slightly more variable than for 1:1 matching.

Figs 6 and 7 show the results of the CPs for the confidence intervals. First, regarding the model-based CI results, in many situations the CP exceeds 95% and the performance appears to be high. However, when the number of events was small, e.g., $p_g = 0.1$ and $p_e = 0.1$, the CI for caliper matching was overly conservative, with a proportion of highly unstable estimates (regression coefficients SE > 100) exceeding 30%, regardless of the matching order. For W_ATT and W_ATTt, CP for model-based CIs was lower when the proportion of the treatment group was high, regardless of the degree of bias. For W_ATO, model-based CI was generally rather conservative. Second, for robust CIs, the CP remained at the level of nominal in the absence of bias, except when the event incidence was 10%, and the estimation was rarely significantly unstable in all situations. This was also the case with the weighting method. The CP of the M_Med and M_Med2 methods tended to be conservative compared to the other matching orders. These trends in model-based and robust CI were generally similar in other conditions.

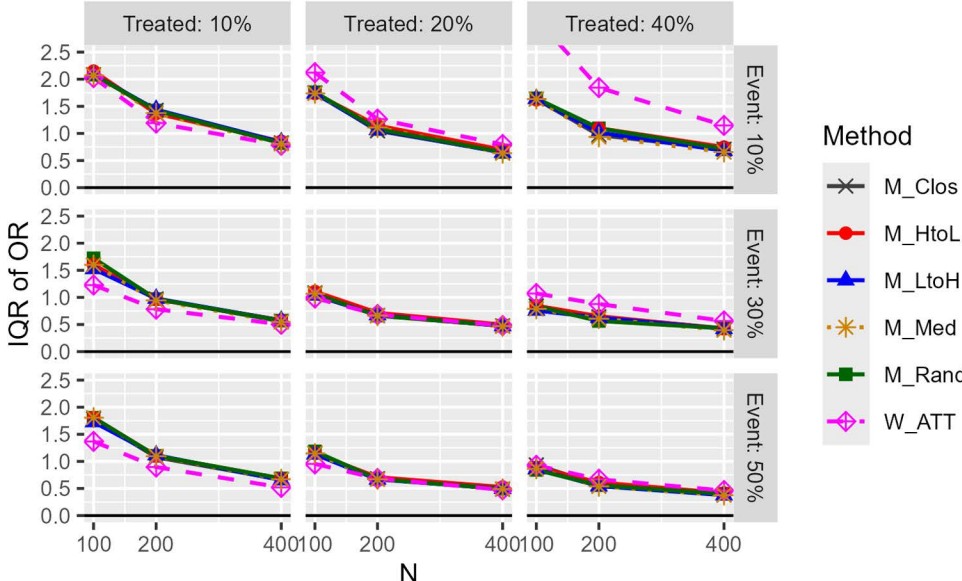

**Fig 4. IQR of estimated odds ratios (unimodal continuous covariate, c statistic: 0.85, true OR: 1.0, matching ratio: 1:1).**

SE results are presented only in the Supplementary Material, as they closely resembled the CP results. The model-based SE exhibited substantial upward bias and is therefore generally not recommended. In contrast, the robust SE showed downward bias when the expected number of events was small. Notably, this bias occurred even when the CP was close to the nominal level, which can be partially attributed to the skewed distribution of the log OR under low event expectation.

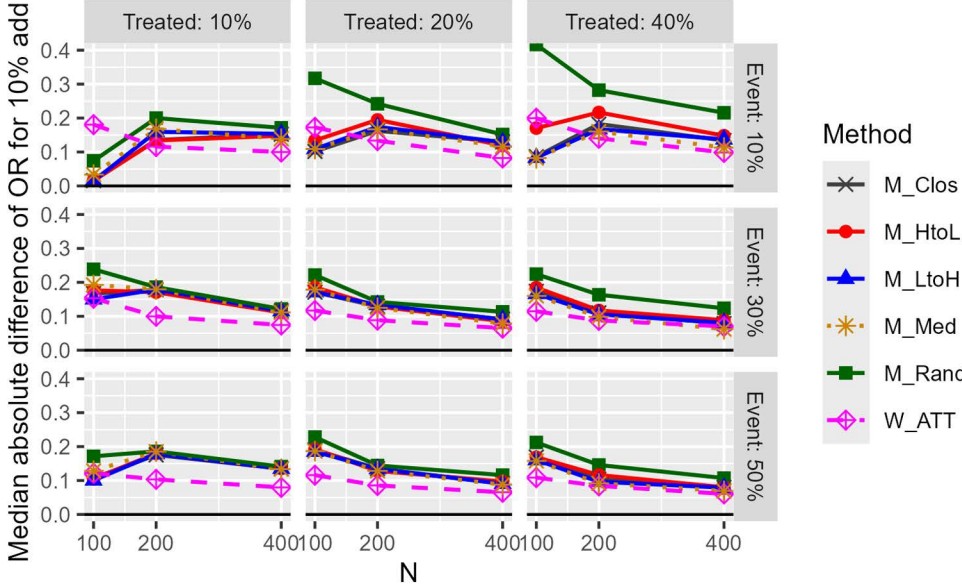

**Fig 5. Median absolute difference of estimated odds ratios for 10% data addition (unimodal continuous covariate, c statistic: 0.85, true OR: 1.0, matching ratio: 1:1).**

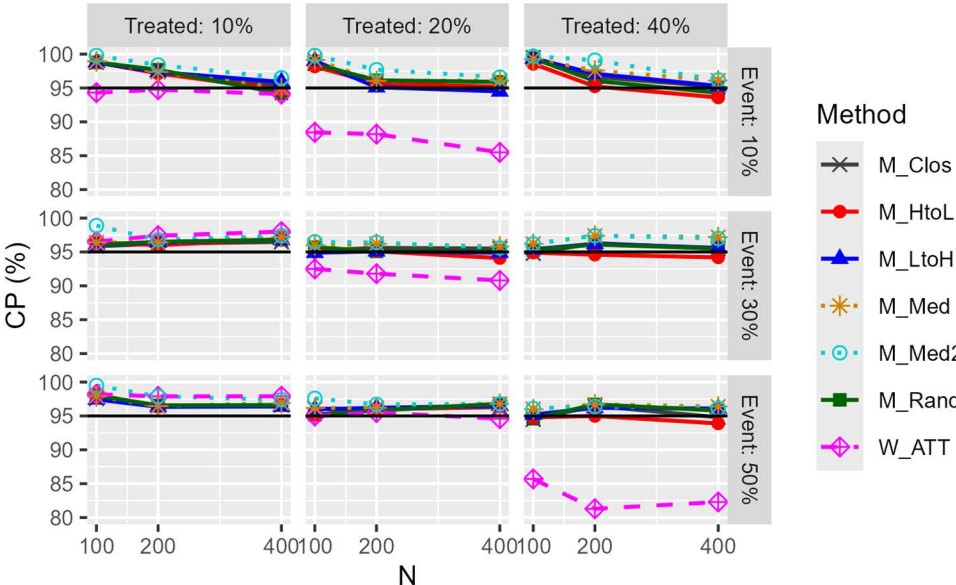

**Fig 6. Coverage probability (CP) of model-based confidence intervals for odds ratios (unimodal continuous covariate, c statistic: 0.85, true OR: 1.0, matching ratio: 1:1).**

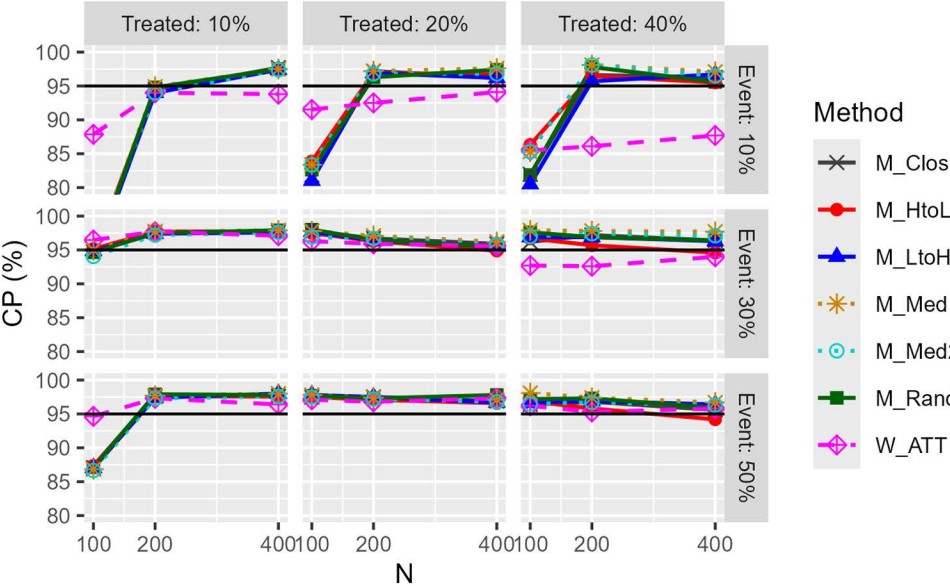

**Fig 7. Coverage probability (CP) of robust confidence intervals for odds ratios (unimodal continuous covariate, c statistic: 0.85, true OR: 1.0, matching ratio: 1:1).**

## 4. Case study

We focused on data obtained from the German Breast Cancer Study (GBCS) Group [19]. In the main study, 720 patients with primary node positive breast cancer were recruited between July 1984, and December 1989. This study used data from 686 subjects with complete data on the covariates [20].

The treatment group was defined as hormone-treated or not (proportion of hormone-treated patients: 35.9%). The aim of this case study is to assess the instability of random order caliper matching under conditions that cover several common situations, similar to the simulation setting, in the context of estimating the effects of hormone therapy. We focused on this dataset because: 1) it is non-randomized clinical trial data for estimating the effect of a treatment intervention; 2) although survival time was originally used as an outcome, by converting it to a binary outcome of survival to a certain time point, we can examine multiple binary outcomes with different incidence rates; 3) it is of moderate sample size (686), and by using a subsample of this data, we can examine the performance of random-order caliper matching on small-sample data of our interest with multiple sample sizes.

The original outcome was overall survival time (24.9% mortality proportion); however, this study assessed mortality at 2 years and 4 years. The incidence of each outcome was 10.0% and 20.2%, respectively. From the original dataset, $N = 100, \ 200$, and $400$ subjects were selected from the top to bottom of the subject ID, and each sample was analyzed. Covariates used to estimate propensity scores were age, tumor size, tumor grade, number of nodes, number of progesterone receptors, and number of estrogen receptors. Using the estimated propensity score, we estimated the odds ratio with greedy nearest neighbor matching with a caliper for each sample. The procedures for the matching order were random (Rand), highest to lowest (HtoL), lowest to highest (LtoH), and in order of the distance between units (Clos). Furthermore, we added $0.1N$ (10%) of data to each sample and calculated the difference between the odds ratio of the merged and original data. In the Rand method, the distribution was presented in 1,000 iterations.

Fig 8 shows the histogram of the estimated propensity score for the original data. The c statistic of the propensity score model was 0.67, which was in the middle of the values set in the simulation study (0.6 and 0.85). Fig 9 illustrates

the odds ratios estimated for each condition. In the condition of $N = 100$ and death by 2 years (upper left panel), as in the simulation, no variation was observed in the random order owing to few events, which limited the pattern of the results. For other conditions, the results varied between 0.2 and 0.3. The other methods also showed some degree of variation in the results. The median value of the M_Rand method did not differ significantly from each of the other three methods, suggesting the stability of the estimates Fig 10 shows the change in odds ratio with the addition of 10% data in each

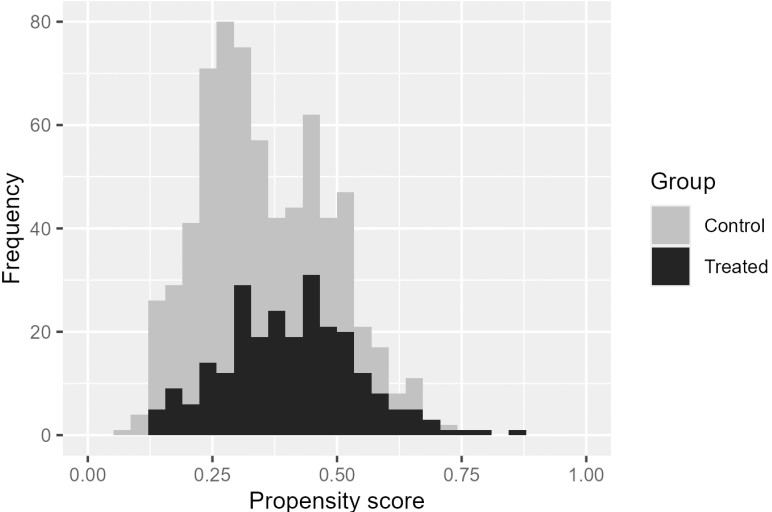

**Fig 8. Histogram of estimated propensity score for GBCS data analyses.**

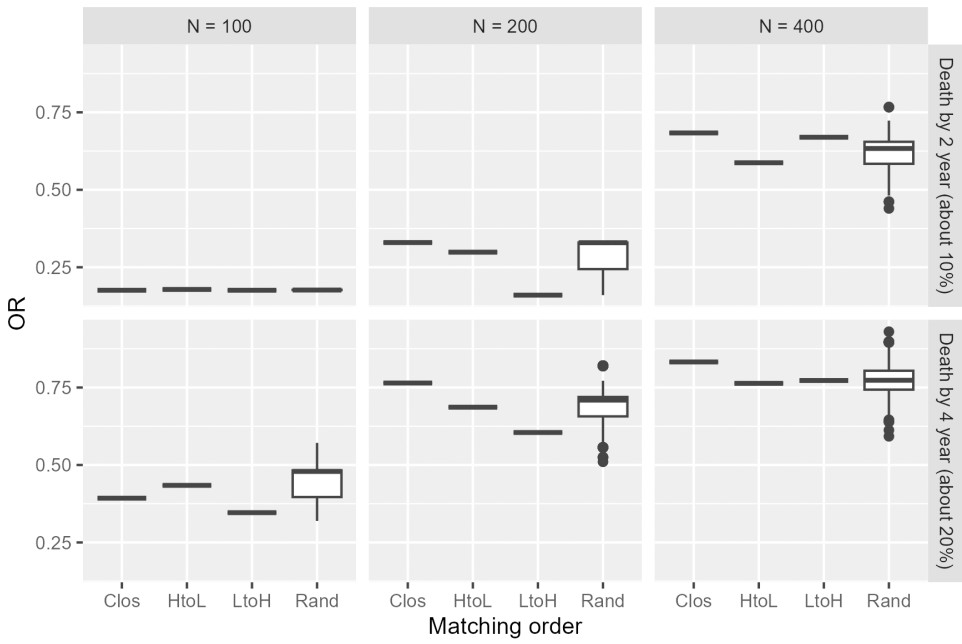

**Fig 9. Odds ratio for GBCS data analyses; point estimates for Clos, HtoL, and LtoH and box-whisker plot of the odds ratios for 1,000 replications of Rand matching.**

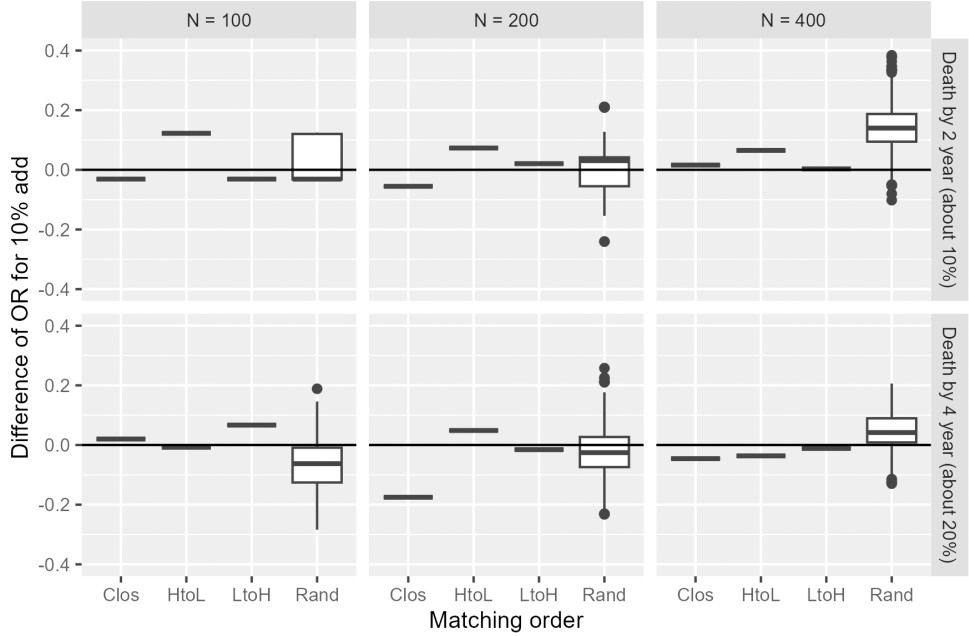

**Fig 10. Difference of estimated odds ratios of 10% data addition for GBCS data analyses; point estimates for Clos, HtoL, and LtoH and box-whisker plot of the odds ratios for 1,000 replications of Rand matching.**

condition. Similar to the simulation results, the risk of a large change due to data addition was higher for the Rand method. Specifically, the impact of data accumulation on the odds ratio was small for all methods except the Rand method, with variations generally remaining close to zero and not exceeding ±0.2. In contrast, the Rand method exhibited a comparatively larger fluctuation, with changes reaching up to approximately 0.4.

## 5. Discussion

We investigated the instability problem of random order matching based on simulations and a case study. From simulation studies, instability might increase when the sample size is small, the true odds ratio is large, the proportion of subjects in the treatment group is large, and the c-statistic for the propensity score model is large. The magnitude of instability may cause cherry-picking of the analysis results and compromise their reproducibility. The simulation results also suggest that random order matching might produce excessive variability in the results when multiple analyses are performed as the data accumulate. The case study results also supported these findings.

From the viewpoint of instability, random order matching should not be used; instead, other order methods should be considered. Among these, the performance of the M_HtoL method is slightly lower, so the other methods may be better. Because the M_HtoL method is the default in the MatchIt package, it should be used with caution. Another option is to perform multiple random order matching (e.g., 100 times) and report the median value as the result. This approach has the advantage of better performance regarding bias and differences between analyses owing to data addition. The M_Med and M_Med2 methods, both based on robust variance, demonstrated comparable performance for inference in this option. However, M_Med2 may be preferable due to its stronger theoretical justification. In addition, regardless of whether this method is used, instability in any given data can be easily evaluated by performing multiple random order matching runs; therefore, it is recommended to check for instability problems. To avoid the instability problem, weighted methods, particularly the overlap weighting method, can be used from a performance perspective (see Supplemental material). The weighting methods may also avoid the problem of subject information being lost that may occur during

matching. In addition, whenever possible, the matching order method should be pre-specified, or at least cherry-picking the matching order from the results should be avoided. For confidence intervals, robust CI is recommended because CP is not lower than the nominal level except for overly sparse situations and/or when the effect estimator has bias, and it is also not overly conservative. The median of 100 random matching iterations was provided in our simulations. This number was chosen because of the computational load in the simulations where many conditions and 1000 number of simulations were set; however, by analogy with the performance of the multiple imputation method, it should provide results with sufficient precision. However, in actual data analysis, the computational load of repetition would be almost negligible. For example, 1,000 or 10,000 iterations will not take too much time if the sample size is not too large.

The limitations of this study are as follows: It is noteworthy that the simulation settings were limited. Although this study focused on binary outcomes, similar problems would likely arise for continuous and survival outcomes. Similar results are expected, particularly for ratio indices such as hazard ratios. Future studies could explore other outcomes using similar investigations. Although the simulations in this study have been performed in situations where the number of expected events is considerably low, there are other methods specialized for sparse situations (e.g., see Gosho et al. [21]). The evaluation of these methods in combination with matching schemes is a subject for future research. Furthermore, situations in which unmeasured confounding factors exist were not investigated in this study. While unmeasured confounders are known to introduce bias in the estimation of treatment effects, their impact on the instability remains unclear. As precision can be enhanced by adjusting for confounders correlated with the outcomes, the presence of unmeasured confounders may lead to increased instability. Although the issue of multiplicity in statistical testing was not addressed in the present study, it is important to recognize the risks associated with conducting multiple statistical tests across repeated analyses with additional data, particularly in the context of confirmatory clinical studies. We focused exclusively on simple propensity score models and matching procedures commonly used by general statistical practitioners. The evaluation for more advanced procedures—such as propensity score or outcome models incorporating complex structures (e.g., interaction terms), kernel-based matching, and machine learning approaches—remains a subject for future research.

## Supporting information

**S1 File. Simulation results for matching with caliper width of 25%, matching without caliper, weighting methods.** (PDF)

**S2 File. Simulation results for matching with caliper width of 15%.** (PDF)

## Author contributions

**Conceptualization:** Kazushi Maruo.

**Data curation:** Kazushi Maruo.

**Formal analysis:** Kazushi Maruo.

**Funding acquisition:** Masahiko Gosho.

**Investigation:** Kazushi Maruo.

**Methodology:** Kazushi Maruo.

**Project administration:** Kazushi Maruo.

**Resources:** Kazushi Maruo.

**Software:** Kazushi Maruo.

**Supervision:** Yusuke Yamaguchi, Ryota Ishii, Masahiko Gosho.

**Validation:** Yusuke Yamaguchi, Ryota Ishii.

**Visualization:** Kazushi Maruo.

**Writing – original draft:** Kazushi Maruo.

**Writing – review & editing:** Yusuke Yamaguchi, Ryota Ishii, Masahiko Gosho.

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
