## [Decision Letter · Decision Letter 0]

20 Aug 2024

PONE-D-24-14920Instability of Estimation Results Based on Caliper Matching with Propensity ScoresPLOS ONE

Dear Dr. Maruo,

Thank you for submitting your manuscript to PLOS ONE. After careful consideration, we feel that it has merit but does not fully meet PLOS ONE’s publication criteria as it currently stands. Therefore, we invite you to submit a revised version of the manuscript that addresses the points raised during the review process.

We look forward to receiving your revised manuscript.

Kind regards,

Md. Belal Hossain, PhD(c)

Academic Editor

PLOS ONE

Journal Requirements:

"JSPS KAKENHI Grant Number JP22K19682"

4. Please expand the acronym “JSPS” (as indicated in your financial disclosure) so that it states the name of your funders in full.

"This work was supported by JSPS KAKENHI Grant Number JP22K19682."

"JSPS KAKENHI Grant Number JP22K19682"

**Additional Editor Comments:**

What if you repeat the analysis multiple times and pool the estimate? This idea is analogous to the analysis technique in multiple imputation literature. Repeating the analysis multiple times and pooling the estimate could solve the instability of the problem.  When comparing the methods, the authors should explain why they used a non-collapsible effect measure. Additionally, methods should be compared in terms of bias, model-based standard error (SE), empirical SE, and coverage probability.The simulation settings are very unrealistic. Please explain why specific parameter values (e.g., c-statistic, odds ratios) were chosen and how they relate to real-world scenarios. It would be best to use an established simulation setting.In continuation of my previous comment, it seems unrealistic to have sample sizes of 100 and 200. It would be desirable to have at least one simulation scenario with parameters similar to those of an established simulation setting. I am backing this approach because an overfitted PS model could lead to a biased effect estimate and may not be suitable for comparing methods in a simulation study. A better approach could be to consider a well-calibrated model, ensuring that there is at least one scenario where all methods are comparable. On top, given the small sample size with or without a rare event, it is surprising that the authors did not consider 1:m or full matching. 1:1 matching is ideally not recommended with a rare exposure or a rare event.The reason for considering 10,000 simulations instead of 1,000 was unclear. Also, how did the authors come up with a repeated sample of 100 to calculate the median?It is not clearly explained why the median was chosen as a summary measure, as calculating standard error for the median is always complex.It was not clear how small additions and/or changes in data worked in the case study.Also, the explanation for the case study is insufficient. Please provide a brief rationale in the introduction section for selecting the low-birth-weight data, how it exemplifies the instability problem, and the motivation for the proposed comparisons based on the case study dataset.There should be a discussion on the case study results, particularly how they compare to the findings from the simulations.Limitations should be discussed such as unmeasured confounding, and other issues in observational data analyses.

Reviewers' comments:

Reviewer's Responses to Questions

**Comments to the Author**

1. Is the manuscript technically sound, and do the data support the conclusions?

Reviewer #1: Yes

Reviewer #2: Yes

2. Has the statistical analysis been performed appropriately and rigorously? 

Reviewer #1: No

Reviewer #2: Yes

3. Have the authors made all data underlying the findings in their manuscript fully available?

Reviewer #1: No

Reviewer #2: Yes

4. Is the manuscript presented in an intelligible fashion and written in standard English?

Reviewer #1: Yes

Reviewer #2: Yes

5. Review Comments to the Author

Reviewer #1: Overall, the paper is crisp and the problem nicely stated and elaborated. I only have severe concerns with the chosen data example, which is exactly the kind of study where propensity methods should not be used. At some places it seems that the translation into English failed (e.g. Line 263 "paired data that return the median" what does it mean?). See my specific concerns below.

Introduction: The authors should give a better motivation for propensity score caliper matching (as opposed to other methods that are less dependent on a stochastic algorithm such as covariate adjustment for propensity score or inverse probability of received treatment weighting). If there are better methods around, why should we take care of another deficiency of PS matching that does not exist with covariate adjustment or IPTW?

Line 47: the instability problem may not be visible, but different researchers may set different seeds which leads to different results despite full prespecification even if they used the same data set. Maybe this could be mentioned here.

Line 114: consider rewriting the sentence "As a modified version of W ATT, a method that excludes more than the 95th percentile of the distribution of propensity scores was also applied (W_ATTt) (Lee et al.)" Does it mean that subjects with propensity scores greater than the 95th percentile were excluded?

Line 185: what does it mean '... except when the number of matched pairs was covariates'?

Lines 208ff Case study. While I like the exposition and the simulation study, I don't think it is wise to use propensity score matching for a risk factor. Moreover, in this example the exact causal ordering of the measured variables is not known (it is unclear if the 'pretreatment' condition is fulfilled for all considered confounding variables). The authors should consider showing an example with a proper intervention variable. If the paper is published, other authors will apply PS matching to similar questions and we should generally discourage from using PS methods with non-interventions, in particular if standard assumptions for causal inference are in doubt. A more proper example can be found, e.g., in the paper of Luijken et al (2024, Biometrical Journal) and the accompanying GitHub repository in the folder https://github.com/KLuijken/CI_CovSel/tree/master/rcode/add-ons/simulated_CABG_example. These authors emulated the data of a real observational study with an intervention using R code, which can be found on the GitHub repository.

Line 219: " the odds ratio for smoking variables ..." I assume the authors mean the odds ratio of smoking vs. not smoking. But anyway, I strongly propose to change the example.

Lines 241ff: I don't quite agree with the conclusion that any of the methods (HtL or LtH) should be recommended. In fact, matching is still arbitrary and not 'wrong' if performed with a different ordering. Hence, as already stated in my comment on line 47, even if the matching algorithm becomes deterministic when using a predefined ordering, it is still one of several possible matchings that are all in line with the caliper requirement. There is no scientific reason to prefer HtL to LtH.

Line 246 & 260: When the authors recommend to use multiple random order matching and report the median, did they investigate the coverage rates of 95% confidence intervals based on the median OR? I would assume that there is some undercoverage if one picks a matching and ignores the variability of matching. Hence I would be more cautious than saying 'there seems to be no major problem' (line 262). The authors could easily add an evaluation of coverage rates at least for the methods which have (almost) no bias. Or they should tone down their conclusion.

The major problem with matching is the number of subjects that cannot be matched, which are lost from the analysis. Hence, one should probably better using weighting or covariate adjustment in practice in order not to throw away precious information.

Figure S4: there are no lines for OR=0.5 and OR=0.75. Please check or add explanation on overlaying curves.

Reviewer #2: This article addresses the implications of the criteria chosen for matching individuals when performing caliper matching with propensity scores in observational studies. The authors expose the current issues regarding reproducibility and instability according to literature and do a large simulation study where they compare different criterias for caliper matching and also compare them to propensity score weighting. The authors also apply these techniques in a case study using publicly available data. This is a relevant work, given that propensity scores in general and matching in particular are widely used in research, and it is also well conducted and presented by its authors. However, I have several concerns regarding the manuscript which are described below:

1. In the Introduction section, the authors mention that cherry-picking is easy to conduct in this type of matching. Is this problem very extended in the literature? Do the authors have access to any bibliography regarding the cherry-picking issue in this type of studies, and/or how effective can it be? If so, I would recommend to include it in this section to give a measure of the magnitude of the problem.

2. The greedy nearest neighbor matching technique should be described in more detail in Section 2, apart from introducing the four types of sorting available in literature.

3. Page 6, lines 102-103: “multinomial distribution with 10 categories” What were the probabilities for each of the 10 categories in this distribution? Were they uniform or unequal?

4. Page 7, lines 106-107: “represented a case where the propensity in treatment selection was close to random or where there was remarkable propensity” Respectively? 0.65 in the c-statistic is close to random and 0.85 remarkable propensity? Please clarify.

5. I understand that the simulation has been done considering that the outcome variable is binary. Would the results be different if the outcome variable was continuous? I think this point should be at least discussed in the manuscript, as continuous outcomes are relatively common in observational studies.

6. Page 9, lines 153-155: “We added 0.1N (10%) of the data to each simulation, and the absolute difference between the odds ratio of the added merged data and that of the original data and the median of the absolute difference were calculated.” Could the authors clarify in the text what was the purpose of this procedure?

7. I think the graphics in the Results section (and the Supplementary Material) would benefit from redefining the limits of the Y axis. I find that, in many cases, the span of the axis is very wide and that makes the lines to be too “crushed” and barely distinguishable from each other, and this could be alleviated by readjusting the amplitude of the Y axis. I would recommend trying to redo the graphics if the authors are able to do so.

8. Page 11, lines 178-180: “For the weighting based analysis method, only the results of the ATT weight-based method are presented in the text because it is one of the most frequently used methods.” I find that W_ATO and W_ATTt showed good performance according to the graphics presented in the supplementary file. Could this statement be revised?

9. I think the authors should include more details about the data used and its origins in the Case Study section, even if it is a dataset taken from an R package. Knowing more details would help the reader to know the quality of the observational study conducted and the expectations they should have regarding propensity adjustments.

10. Is there any possibility to assess the actual bias of the estimations from the case study? Are the authors aware of any data on real randomized experiments on this matter?

Typos:

11. Page 6, line 99: “p gj” I think it should read as “p_{gj}”

12. Page 14, lines 233-235: the verb tense should be changed in this sentence (i.e. every “was” should be replaced by “is”).

6. PLOS authors have the option to publish the peer review history of their article (what does this mean? ). If published, this will include your full peer review and any attached files.

**Do you want your identity to be public for this peer review?** For information about this choice, including consent withdrawal, please see our Privacy Policy .

Reviewer #1: **Yes: ** Georg Heinze

Reviewer #2: No

---

## [Decision Letter · Decision Letter 1]

15 Jan 2025

PONE-D-24-14920R1Instability of Estimation Results Based on Caliper Matching with Propensity ScoresPLOS ONE

Dear Dr. <!--StartFragmentMaruo,

Thank you for submitting your manuscript to PLOS ONE. After careful consideration, we feel that it has merit but does not fully meet PLOS ONE’s publication criteria as it currently stands. Therefore, we invite you to submit a revised version of the manuscript that addresses the points raised during the review process.

We look forward to receiving your revised manuscript.

Kind regards,

Academic Editor

PLOS ONE

Additional Editor Comments:

The authors made significant changes in the revised version. However, two major issues still need to be addressed before publication.

• Pooling the estimate: It was proposed to pool the estimate and calculate SE to reflect the improved uncertainty due to polling. There should be at least one comparison of how the proposed method works compared to pooling the estimate. Here is an example where the authors used the modified multiple outputation technique to account for within and between variations in the estimate for improving precision (doi: 10.1007/s12561-024-09461-6).

• The authors should consider a collapsible effect measure and compare the methods in terms of bias, model-based standard error (SE), empirical SE, and coverage probability. The justification for not doing these was insufficient and unreasonable (doi: 10.1002/sim.8086).

Reviewers' comments:

Reviewer's Responses to Questions

**Comments to the Author**

1. If the authors have adequately addressed your comments raised in a previous round of review and you feel that this manuscript is now acceptable for publication, you may indicate that here to bypass the “Comments to the Author” section, enter your conflict of interest statement in the “Confidential to Editor” section, and submit your "Accept" recommendation.

Reviewer #2: All comments have been addressed

Reviewer #3: (No Response)

Reviewer #4: (No Response)

2. Is the manuscript technically sound, and do the data support the conclusions?

Reviewer #2: Yes

Reviewer #3: Yes

Reviewer #4: Yes

3. Has the statistical analysis been performed appropriately and rigorously? 

Reviewer #2: Yes

Reviewer #3: Yes

Reviewer #4: Yes

4. Have the authors made all data underlying the findings in their manuscript fully available?

Reviewer #2: Yes

Reviewer #3: Yes

Reviewer #4: Yes

5. Is the manuscript presented in an intelligible fashion and written in standard English?

Reviewer #2: Yes

Reviewer #3: Yes

Reviewer #4: Yes

6. Review Comments to the Author

Reviewer #2: The authors have successfully addressed all the comments posed in the first revision, hence I recommend the manuscript to be accepted for publication.

Reviewer #3: Thank you for the opportunity to review this manuscript. This study investigates the effect of matching order in propensity score matching to demonstrate the potential instability of results, particularly in small-to-medium sample sizes. Overall, I believe this is a well conducted study that raises an interesting point about the potential for cherry-picking results in such studies. Below are two specific comments, and questions that I had while reviewing this study:

1. line 160: "the expected number of events was one for each group" is this a realistic scenario for any type of analysis? I am struggling to think of an area of study where we can do any serious sort of analysis with a single event, especially in the context of a frequentist statistical framework. In situations like this I would think that the matching order of the propensity scores is pretty far down on the list of major issues. I would think that this is more than a very small information setting, to the point where the issues this paper are discussing almost become moot due to the severity of the violation of other statistical assumptions. I think maybe a bit more explanation regarding the explicit purpose of modeling such a scenario may help readers better understand the purpose of using such an extreme simulation setting.

2. At several points in the manuscript the idea of conducting multiple analyses as more data are accumulated is discussed. This is an interesting area that is receiving more attention recently in the clinical trial literature (adaptive clinical trials). My concern with this is the potential for the potential inflation of Type I error and cherry-picking results/p-hacking that can arise from this stream of methodological choices if not conducted carefully and transparently reported (preferably a priori). While this is more of a practical concern not directly relevant to the simulation per se (as the true effects are known) I think it may be worthy of adding some language surrounding this concept cautioning the potential issues surrounding this approach.

Reviewer #4: Introduction section:

•The aim could be stated more succinctly at the beginning to ensure the reader understands the problem immediately.

•The introduction could benefit from a brief discussion of how the findings will address real-world applications or improve current practices.

•The introduction could briefly outline why the chosen design (simulations and case studies) is most suitable for evaluating instability.

•Briefly justify the choice of simulation studies as the primary method to explore instability and validate findings through case studies.

•Revise for conciseness and grammatical clarity to improve readability. Some sentences are verbose or contain minor grammatical inconsistencies, such as "especially, particularly for a non-large sample" and "no research for variability in propensity score analysis due owing to multiple analyses."

Greedy nearest neighbor matching section:

•Quantify or provide examples of how variability due to data accumulation affects Methods 1-3 compared to random order matching.

•Including following considerations in discussion section would provide a more balanced perspective:

1. The assumption of a unimodal propensity score distribution might not always hold in real-world data.

2. The exclusion of interaction effects or non-linear covariate relationships in the propensity score model could limit applicability to more complex datasets.

Simulation design section:

•The unimodal distribution assumption for propensity scores may not always reflect real-world scenarios where multimodal distributions or non-normality are observed. A sensitivity analysis incorporating such variations could strengthen the conclusions.

•While different matching orders are evaluated, the robustness of caliper matching to outliers or extreme propensity scores could be explored further. For example, alternative caliper widths, kernel-based matching, or machine learning approaches for propensity score estimation might provide additional insights.

Case study section:

•“The risk of a large change due to data addition was higher for the Rand method.” Quantify “large change” to contextualize this risk.

•For the low birth weight data, the authors focus on a binary outcome and a single matching ratio. In contrast, the GBCS data are analyzed with different outcomes, sample sizes, and ratios. While this demonstrates versatility, the rationale for the differing approaches between datasets is not discussed. Clarifying this would improve coherence.

•The c-statistic for the propensity score model in the low birth weight data is not reported. Given its importance in propensity score analyses, this omission should be rectified for consistency with the GBCS data discussion.

•The impact of adding 10% data on odds ratios is presented but not fully contextualized. For instance: Does the larger change in the Rand method suggest a lack of robustness, or is it expected behavior given its design? How does this finding translate into recommendations for researchers dealing with incomplete or expanding datasets?

Discussion section:

•The recommendation to use 1,000 iterations for random order matching (instead of 100) is presented without sufficient justification. A brief discussion of how this choice balances computational load and result precision would strengthen this recommendation.

•The discussion of unmeasured confounders is minimal. It would benefit from a brief exploration of how such confounding might exacerbate instability or bias results.

7. PLOS authors have the option to publish the peer review history of their article (what does this mean? ). If published, this will include your full peer review and any attached files.

**Do you want your identity to be public for this peer review?** For information about this choice, including consent withdrawal, please see our Privacy Policy .

Reviewer #2: No

Reviewer #3: No

Reviewer #4: No

---

## [Author Response · Author response to Decision Letter 2]

25 Apr 2025

This is uploaded separately as a Word file.

---

## [Decision Letter · Decision Letter 2]

12 May 2025

Instability of Estimation Results Based on Caliper Matching with Propensity Scores

PONE-D-24-14920R2

Dear Dr. Maruo,

We’re pleased to inform you that your manuscript has been judged scientifically suitable for publication and will be formally accepted for publication once it meets all outstanding technical requirements.

Kind regards,

Belal Hossain, PhD

Academic Editor

PLOS ONE

Additional Editor Comments (optional):

Reviewers' comments:

Reviewer's Responses to Questions

**Comments to the Author**

1. If the authors have adequately addressed your comments raised in a previous round of review and you feel that this manuscript is now acceptable for publication, you may indicate that here to bypass the “Comments to the Author” section, enter your conflict of interest statement in the “Confidential to Editor” section, and submit your "Accept" recommendation.

Reviewer #3: All comments have been addressed

Reviewer #4: All comments have been addressed

2. Is the manuscript technically sound, and do the data support the conclusions?

Reviewer #3: Yes

Reviewer #4: Yes

3. Has the statistical analysis been performed appropriately and rigorously? 

Reviewer #3: Yes

Reviewer #4: Yes

4. Have the authors made all data underlying the findings in their manuscript fully available?

Reviewer #3: Yes

Reviewer #4: Yes

5. Is the manuscript presented in an intelligible fashion and written in standard English?

Reviewer #3: Yes

Reviewer #4: Yes

6. Review Comments to the Author

Reviewer #3: The authors have addressed all my comments. Thank you for the opportunity to review this interesting, and insightful piece of work.

Reviewer #4: (No Response)

7. PLOS authors have the option to publish the peer review history of their article (what does this mean? ). If published, this will include your full peer review and any attached files.

**Do you want your identity to be public for this peer review?** For information about this choice, including consent withdrawal, please see our Privacy Policy .

Reviewer #3: No

Reviewer #4: No

---

## [Editor Report · Acceptance letter]

PONE-D-24-14920R2

PLOS ONE

Dear Dr. Maruo,

I'm pleased to inform you that your manuscript has been deemed suitable for publication in PLOS ONE. Congratulations! Your manuscript is now being handed over to our production team.

Kind regards,

on behalf of

Dr. Md. Belal Hossain

Academic Editor

PLOS ONE